# Revealing the aging process of solid electrolyte interphase on SiO$_x$ anode

Guoyu Qian [1,2,11], Yiwei Li [1,11], Haibiao Chen [1,3], Lin Xie [4], Tongchao Liu[5], Ni Yang[1], Yongli Song[1], Cong Lin [1,6], Junfang Cheng[7,8], Naotoshi Nakashima[7], Meng Zhang[9], Zikun Li[9], Wenguang Zhao[1], Xiangjie Yang[2], Hai Lin[1], Xia Lu [2], Luyi Yang [1] ✉, Hong Li [10], Khalil Amine [5], Liquan Chen[10] & Feng Pan [1] ✉

As one of the most promising alternatives to graphite negative electrodes, silicon oxide (SiO$_x$) has been hindered by its fast capacity fading. Solid electrolyte interphase (SEI) aging on silicon SiO$_x$ has been recognized as the most critical yet least understood facet. Herein, leveraging 3D focused ion beam-scanning electron microscopy (FIB-SEM) tomographic imaging, we reveal an exceptionally characteristic SEI microstructure with an incompact inner region and a dense outer region, which overturns the prevailing belief that SEIs are homogeneous structure and reveals the SEI evolution process. Through combining nanoprobe and electron energy loss spectroscopy (EELS), it is also discovered that the electronic conductivity of thick SEI relies on the percolation network within composed of conductive agents (e.g., carbon black particles), which are embedded into the SEI upon its growth. Therefore, the free growth of SEI will gradually attenuate this electron percolation network, thereby causing capacity decay of SiO$_x$. Based on these findings, a proof-of-concept strategy is adopted to mechanically restrict the SEI growth via applying a confining layer on top of the electrode. Through shedding light on the fundamental understanding of SEI aging for SiO$_x$ anodes, this work could potentially inspire viable improving strategies in the future.

During the past decades, the pursuit of high-energy-density Li-ion batteries (LIBs) has stimulated the development of high-capacity negative electrode (anode) materials[1–3]. Among a variety of competitive candidates, silicon (Si) anodes based on alloying reactions have drawn worldwide interests from both academia and industry due to the higher specific capacity (~4200 mAh g$^{-1}$) compared with intercalation-based graphite[4]. However, enabling extra Li$^+$ storage per mass will inevitably bring about a significant volume swing. The volume variation of Si upon lithiation/delithiation could be as high as 300%, causing active material pulverization, electrode disintegration and unstable solid electrolyte interphace (SEI) layer[4–7]. Therefore, silicon oxide (SiO$_x$, 0<x<2) anode has been intensively investigated as a promising alternative for Si as it exhibits much improved volumetric stability (118% volume change) and high theoretical capacity (2680 mAh g$^{-1}$ for SiO)[8–10]. Moreover, the production of SiO$_x$ is industrially scalable by the chemical vapor deposition method, making it a more

[1]School of Advanced Materials, Peking University, Shenzhen Graduate School, Shenzhen, China. [2]School of Materials, Sun Yat-sen University, Shenzhen, China. [3]Institute of Marine Biomedicine, Shenzhen Polytechnic, Shenzhen, China. [4]Department of Physics, Southern University of Science and Technology, Shenzhen, China. [5]Chemical Sciences and Engineering Division, Argonne National Laboratory, Argonne, IL, USA. [6]Department of Applied Biology and Chemical Technology, The Hong Kong Polytechnic University, Hong Kong S.A.R, China. [7]International Institute for Carbon-Neutral Energy Research (WPI-I2CNER), Kyushu University, Fukuoka, Japan. [8]SJTU Paris Elite Institute of Technology, Shanghai Jiao Tong University, Shanghai 200240, P. R. China. [9]BTR New Material Group Co., Ltd, Shenzhen, China. [10]Institute of Physics, Chinese Academy of Sciences, Beijing, China. [11]These authors contributed equally: Guoyu Qian, Yiwei Li. ✉e-mail: yangly@pkusz.edu.cn; panfeng@pkusz.edu.cn

attractive Si-based anode material in the near term[4]. In recent years, tremendous efforts have been made to optimize Si-based anodes, including structural modification[11,12], binder[13,14] and electrolyte[15] design.

For most anodes in LIBs, SEI forms through the electrochemical reduction of the components in the electrolyte, hence, its composition and structure depend on many variables, including salt, solvent, or in many cases, additives[16,17]. As an important component, SEI on graphite anode has been extensively studied for more than 30 years[18,19]. Previously, through combining various in situ characterizations, we have depicted the formation mechanism of SEI on graphite anode[20]. Recently, works by Grey group and Cui group identified that the main components of the SEI in Si anode are $Li_2CO_3$, $RO(CO_3)Li_2$, LiF, $Li_2O$, and polymers, by using solid-state nuclear magnetic resonance spectroscopies (ssNMR)[5] and scanning transmission electron microscopy with electron energy loss spectroscopy (STEM-EELS)[21], respectively. Moreau and co-workers have also employed low-loss EELS to identify SEI species on nano-sized Si anode[22]. It is revealed that the volumetric instability of Si-based anodes leads to "breathing" SEI layers[23], which continuously thicken with cycles, causing the capacity fading of Si-based anodes[24]. To have a clearer view on the aging process of SEI in Si-based anodes, direct observation techniques are highly desirable. Previously, with the aid of energy dispersive X-ray spectroscopy (EDS) mapping, cryogenic scanning transmission electron microscopy (cryo-STEM) and chemo-mechanical modelling techniques, Wang et al. successfully demonstrated the inward growth behaviors of SEI on Si nanowires, which eventually caused the disruption of an electron conducting pathways[25]. However, these above-mentioned techniques are unsuitable for directly observation of the structural evolution of SEI on micro-sized particles with large volume swings, while the traditional microscope-based technique is only suitable for 2D observation of the cross-sectional structure of the electrode without 3D information. In this case, new approaches should be adopted.

In this work, through integrating a series of nanoscale probing and imaging techniques, a concerted approach is proposed for dynamic analysis of SEI evolution on micro-sized commercial $SiO_x$. First, we visualized the growing process of the SEI layer at different stages of cycling and further reconstructed the 3D morphology of SEI. Based on the characteristics of different SEI formed on $SiO_x$ particles, an aging pathway of SEI is proposed. In addition, through directly measuring the electronic conductivity as well as the chemical information in microscale local areas, we speculate the capacity fading is mainly attributed to the free growth of SEI layer, which causes the breakdown of the electron percolating network composed of conductive agents (e.g., carbon black particles) around $SiO_x$ particles and leads to a large IR drop. Inspired by this fundamental understanding, we demonstrated a simple approach to effectively regulate the aging process of SEI to significantly improve the cycling stability of $SiO_x$.

## Results and discussion

### Evolution of SEI thickness on $SiO_x$

To preclude interfering factors, commercial grade $SiO_x$ (x is measured to be 0.68) microparticles with a D50 value of 5.2 μm (Supplementary Fig. 1) without further treatment was chosen as the active material in this work. Both the X-ray diffraction (XRD) pattern and transmission electron microscope (TEM) images (Supplementary Fig. 2) show that pristine $SiO_x$ exists in an amorphous form, which is consistent with previous findings[26,27]. To obtain more statistically reliable electrochemical data, 100 half-cells were assembled using $SiO_x$ as positive electrode materials and cycled at a specific current of 750 mA g$^{-1}$ for 300 cycles. Figure 1a shows a general trend where after an activation process (~ 50 cycles), the capacity of $SiO_x$ cell stabilized around 1170 mAh g$^{-1}$, then a downward inflection point emerged somewhere between 50 and 100 cycles. After 300 cycles, an average capacity retention of 51.8 % is obtained for $SiO_x$. To obtain more detailed information from the galvanostatic cycling data, dQ/dV differential capacity curves of one cell are drawn based on its voltage profiles at different cycle numbers (Fig. 1b). Taking the charging process for instance, at the 10th cycle, well-defined peaks corresponding to different redox reactions of $SiO_x$ can be observed in the dQ/dV curve. Whereas after 50 cycles, the peak intensity around 0.33 V vs Li/Li$^+$ significantly decreased and the two peaks at 0.43 and 0.49 V vs Li/Li$^+$ merged into one broad peak. During the following cycles, the peak attenuation continuously aggravated with cycle numbers, and only two broad peaks could be observed at the 300th cycle, corresponding to the capacity fading tendency shown in Fig. 1a. In addition, the shifting of main peaks towards higher potential as well as the growing charge transfer impedance (Supplementary Fig. 3) suggest the

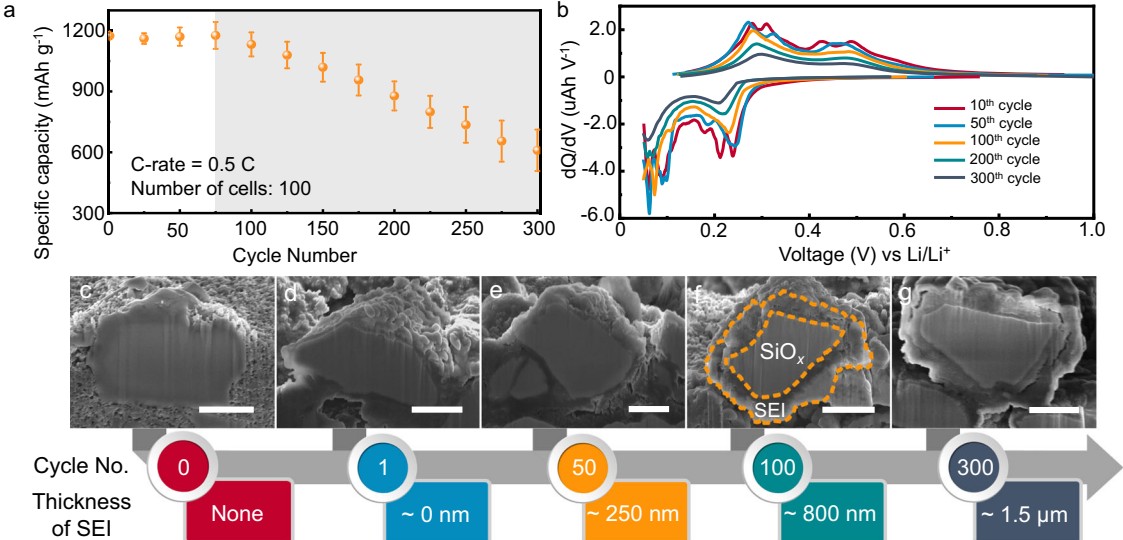

**Fig. 1 | Growth of SEI on $SiO_x$ particles during cycling. a** Cycling performance of 100 $SiO_x$||Li half-cells at the specific current of 750 mA g$^{-1}$. The error bars represent the variation range of the specific capacity of different cells and the dots represent the average specific capacities. **b** dQ/dv curves of a $SiO_x$||Li cell at different cycle numbers. **c–g** Cross-sectional SEM images of a $SiO_x$ particle in the electrode without cycling, after the 1st, 50th, 100th and 300th cycles, respectively. The scale bar = 2 μm.

reaction kinetics become increasingly sluggish. Same conclusions can be drawn from the discharging curves in Fig. 1b based on the similar peak variation trends.

Further experiments have been carried out to pinpoint the origin of electrode deterioration. On the one hand, by applying a lower specific current (150 mA g$^{-1}$) to the half-cell after long-term cycling (150 cycles), a capacity of 1155 mAh g$^{-1}$ could be restored (Supplementary Fig. 4). This result suggests that instead of the degradation of SiO$_x$ or the collapse of the conductive network, the capacity fading during galvanostatic cycling is mainly contributed by the retarded reaction kinetics. On the other hand, electrolyte consumption and Li metal electrode degradation have been previously reported to cause the capacity fading of Si-based electrodes[28]. To preclude those factors, cycled cells were disassembled and reassembled with fresh Li anode and electrolyte. As a result, the capacity of the reassembled half-cell could not be restored after replacing Li and electrolyte (Supplementary Fig. 5). From these results, one could speculate the capacity loss during cycling is mainly caused by an aged SEI that hinders the charge transfer process.

For direct observation of SiO$_x$ particles, electrodes were milled using a focused ion beam (FIB) and the cross-sections of the topmost particles were examined using a scanning electronic microscope (SEM) and an optical microscope. Figure 1c-g show the cross-sectional SEM images a pristine particle, as well as delithiated particles after the 1$^{st}$, 50$^{th}$, 100$^{th}$, and 300$^{th}$ cycle, respectively. After the first delithiation, there appear to be a boundary developed on the surface of this outer layer and the originally loose space between SiO$_x$ particles is filled out (Supplementary Fig. 6). This transition could be attributed to the growth of SEI, as suggested by X-ray photoelectron spectroscopy (XPS) results in Supplementary Fig. 7. The C 1$s$, O 1$s$, F 1$s$ spectra identify the presence of C-C, C-O, C = O, Li-O, and Li-F bonds on the surface of electrodes, which are typical in the SEI layers of Si-based anode materials[14]. Therefore, this outer layer is defined as the effective SEI in a broad sense. Here we measured the thickness evolution of this SEI layer at different cycle numbers. After 50 cycles, the thickness of the SEI layer has grown to approximately 250 nm. After 100 cycles, the

thickness of the layer doubled to 500-800 nm. It is noteworthy that the accelerated thickness growth between the 50$^{th}$ and 100$^{th}$ cycle is synchronized with the capacity fading shown in Fig. 1a, suggesting the possible correlation between cell failure and SEI thickening. After a total of 300 cycles, an SEI as thick as 1.5 µm can be observed on the topmost SiO$_x$ particles of the electrode (more SiO$_x$ particles with aged SEI could be found in Supplementary Fig. 8). Similar SEI can be observed on SiO$_x$ particles with a higher x value (Supplementary Fig. 9).

## Three-dimensional structure of the SEI

To reconstruct a 3D model of the particle and the SEI layer, auto-slicing technique was employed to sequentially slice electrodes under different cycling states. Figure 2a−e and Supplementary Movie 1 demonstrate a complete slicing process of a SiO$_x$ particle on the top of the electrode. As a result, the 3D structure of a single SiO$_x$ particle can be obtained (Fig. 2f−j), where its bulk (purple) can be differentiated from its SEI (cyan). The pristine particle is dense with a smooth surface, with carbon black particles covering the surface. After the first lithiation, an incompact SEI layer could be observed around the particle. Additionally, the interference of beam damage to the sample was studied by comparing samples prepared from FIB and cryo-ultramicrotomy (Supplementary Fig. 10). Although a slightly higher porosity can be observed in the sample prepared by FIB (which can be attributed to the possible beam damage), both samples exhibit SEIs with a loose interior layer and a dense exterior layer, confirming the suitability of this method for observing SEI with considerable thickness (sub-micron level). In comparison, for anode materials with thin and fragile SEI layers (e.g. Si), both Ga$^+$ and electron beams might distort the SEI morphology. Surprisingly, this SEI layer dramatically diminished after the first delithiation. Since freshly formed SEI layers can be fragile and not firmly adhered to SiO$_x$ while the re-oxidation of SEI is very unlikely based on the low initial coulombic efficiency (60%, Supplementary Fig. 11), the SEI thinning process can be attributed to mechanical detachment from the surface of SiO$_x$ particle as it shrank during delithiation. The gradually increasing coulombic efficiency

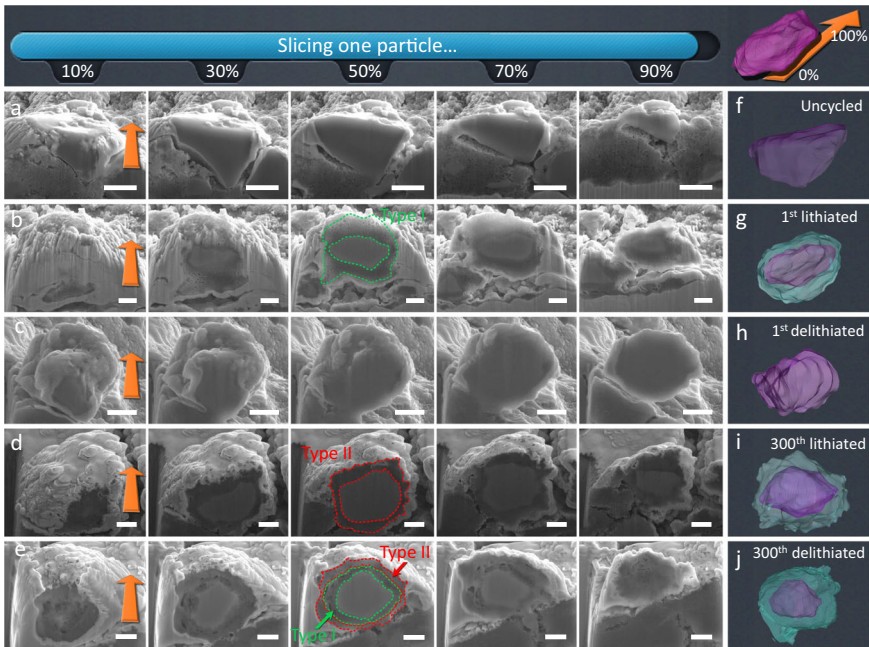

**Fig. 2 | Three-dimensional representation of SiO$_x$ particles and their SEI under different electrochemical status. a−e** Cross-sectional images of a SiO$_x$ particle in the electrode without cycling, after 1$^{st}$, 50$^{th}$, 100$^{th}$ and 300$^{th}$ cycles under a sequential slicing process, respectively. The orange arrow represents the slicing direction. The dynamic process is shown in Supplementary Movie 1. **f−j** The corresponding 3D reconstructed SiO$_x$ particles and their SEI obtained through SEM tomographic microscopy. The scale bar = 2 µm.

during sequential cycles also suggests that after the first cycle SEI residues continue to accumulate on the $SiO_x$ surface until it is completed covered by SEI. By sharp contrast, after the 300th lithiation, a thick SEI layer was developed on the particle, but much denser compared with the SEI observed in Fig. 2b. Interestingly, upon the 300th delithiation, the SEI layer developed into two parts: a loose inner region and a dense outer region. Herein, we designate the loose SEI as Type I SEI and the dense SEI as Type II SEI. A key finding from Fig. 2d and e is that, after long-term cycling, only Type II (dense) SEI was observed on the lithiated $SiO_x$ particle, whereas an SEI layer consisting of Type I inner region and Type II outer region was formed after delithiation. Note that Type I and Type II SEI only differ in the morphology, while they might exhibit very similar chemical compositions.

## Aging process of the SEI

Based on the above observations, we propose a growth mechanism of the SEI on the topmost $SiO_x$ particles as illustrated in Fig. 3. An SEI layer is initially formed on the expanded $SiO_x$ upon the first lithiation; during the first delithiation, the freshly formed Type I SEI cannot shrink at the same pace as the particle due to the lack of adhesion force between them, leading to partial detachment from $SiO_x$ particles (Fig. 3a), which is supported by the re-emerged Si signal in the XPS (Supplementary Fig. 7) and the gradual increase of coulombic efficiency during initial 5 cycles (Supplementary Fig. 11). As the cycle continues, the loosely structured Type I SEI not only grows thicker, but also evolves into the dense Type II SEI, whose morphology remains relatively stable during sequential cycles. In this model, the outer Type II SEI is anchored on the $SiO_x$ particle via the mechanically "flexible" Type I SEI layer (Fig. 3b). Upon delithiation, the Type I SEI is stretched with the shrinkage of $SiO_x$ without causing the structural collapse of Type II SEI; whereas during lithiation, the Type I SEI is compressed between the "dense shell" (Type II SEI) and the expanding $SiO_x$ particle, blurring the boundary between Type I and Type II regions. Therefore, distinctive two-layered SEI structure is only visible under the delithiated state (Fig. 3c). As such accordion-like compress-and-stretch process repeats, the SEI densifies inwardly from its outer layer, instead of undergoing a snowball-like outward growth process. It should be noted that due to the existence

of soft polymeric SEI components, the outer layer can mechanically withstand the volume changes as most of the stress can be buffered by the inner layer. In the end, a thick yet dense Type II SEI is formed on the $SiO_x$ particle, accompanied with capacity decay. In this sense, the aging process of SEI can be regarded as the accumulation of Type II SEI due to the free volume expansion/shrinkage of $SiO_x$ particles during the cycling process. Interestingly, SEI with such thickness has not been observed on Si anodes, which might be attributed to the large volume variation of Si generally leads to particle pulverization and collapse of SEI during its early growing stage, hence no Type II SEI can be formed.

## Conductive network through the SEI region

Conventionally, SEI is recognized as an ionically conductive and electronically insulating layer. For a thin SEI layer, typically in the case of a graphite anode, this statement holds true as electrons can tunnel through the nanometer-thick insulating SEI layer covered on graphite particles. For a micrometer-thick SEI layer fully covering on $SiO_x$ (Supplementary Movie 1), the electron tunneling model can no longer apply since the topmost $SiO_x$ particles remained electrochemically active (as demonstrated in Fig. 2d and e) despite the capacity decay. It can be speculated that there might be a conductive network running through the SEI layer. To understand the conducting behavior of the SEI, nano-probes[29] were employed to directly measure the electronic conductivity on the cross-sectional regions of $SiO_x$ particle, SEI layer and conductive agent (CA) -binder matrix (as shown in Fig. 4a, b and c, respectively). The resistance changes $\frac{dV}{dI}$ with respect to the applied voltage in all three regions are shown in Fig. 4d. The value of intrinsic resistivity for the CA-binder matrix is obtained from the black curve near the zero bias where the concentration of charge carrier has not reached saturation, and the values of $SiO_x$ and SEI are obtained from the curve at a high bias where the contact Schottky Barriers between the probes and the material are overcome[28]. As a result, the measured conductivity of the aged SEI ($7.77×10^{-1}$ S/cm) is lower than that of the CA-binder matrix ($8.61×10^0$ S/cm), but higher than that of bulk $SiO_x$ ($6.75×10^{-2}$ S/cm, see details in Supplementary Fig. 12 and Supplementary Note 1). This result contradicts the common knowledge that the SEI is electronically insulating. Since most compositions in SEI are

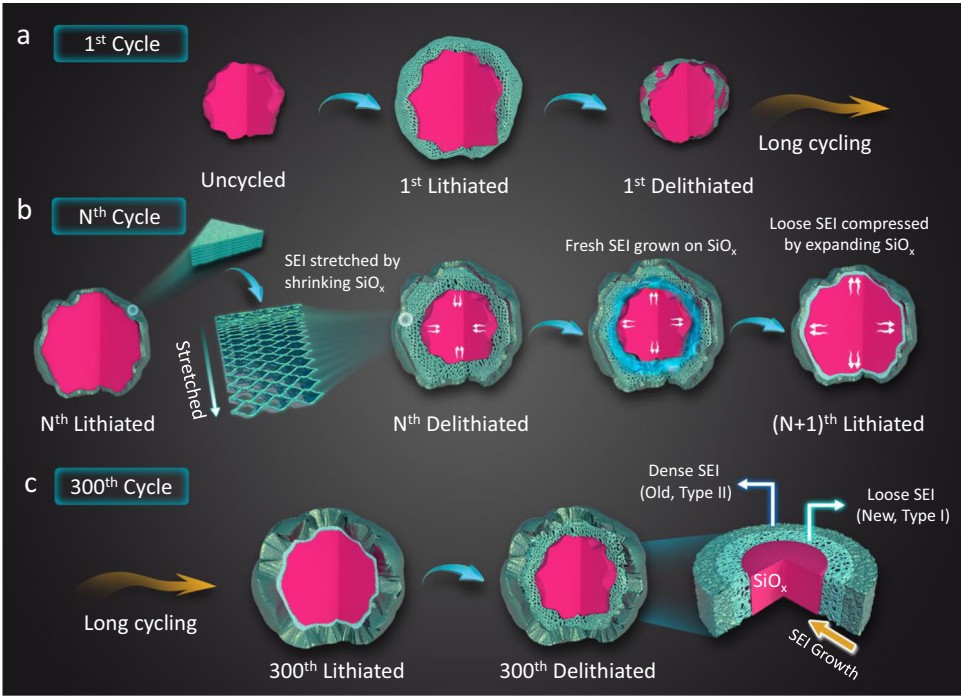

**Fig. 3 | Schematic illustration of the SEI aging process on $SiO_x$.** Structural evolution of SEI formed on $SiO_x$ particle **a** at the initial cycle, **b** during cycling and **c** after 300th cycle.

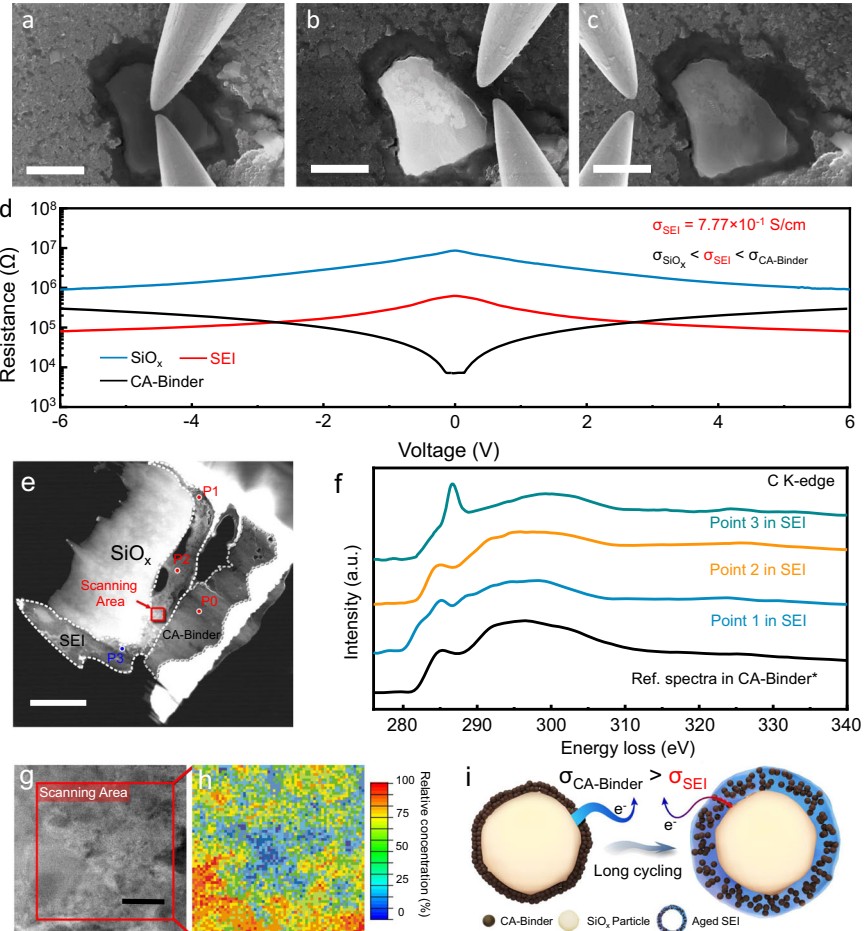

**Fig. 4 | The electronic conductivity and the conductive composition distribution of the SEI region.** Measuring the electronic conductivity of different areas on a section of long-cycled $SiO_x$ electrode: **a** the $SiO_x$ particle; **b** the SEI region; **c** the CA-binder mixture. Scale bar = 2 μm. **d** Resistivity vs. voltage curves in different regions. **e** Electron transparent FIB lift-out lamellae of a $SiO_x$ particle and its SEI with surrounding CA-binder mixture. Scale bar = 2 μm. **f** EELS spectra of the C K-edge from the reference and Point 1-3 (SEI region) in Fig. 4e. **g**, **h** CA-binder distribution of the scanning area in Fig. 4e, obtained from fitting with the correlation C 1s K-edge near edge fine structural peak positions of the reference. Pixels in Fig. 4h represents the planar concentration distribution of conductive agents. Scale bar = 100 nm. **i** The proposed SEI aging mechanism for $SiO_x$ during cycling.

known to be electronically insulating, other conductive agents might be involved in electronic conduction.

As mentioned above, the aging process of SEI is associated with significant layer thickening, which might intrude into the CA-binder domain. Therefore, one could speculate that some CA particles might have been buried during the aging process, facilitating the electronic conductivity of SEI. To directly prove this assumption, EELS was used to analyze different areas of an electron transparent FIB lift-out lamellae, including a $SiO_x$ particle, its SEI and surrounding CA-binder matrix (Fig. 4e). The reference point was sampled from the CA-binder domain in a pristine $SiO_x$ electrode, which exhibits very similar C K-edge spectra with CA-binder domain in the cycled electrode (Supplementary Fig. 13). As shown in Fig. 4f, the spectra of P1 and P2 resemble to the reference, suggesting similar electronic configurations of C in these areas to that of CA-binder mixture, hence the existence of conductive carbon in the SEI. By contrast, P3 exhibits significantly different EELS signal. In this composite electrode, carbon black exhibits a unique sp2 electronic configuration compared with the sp3 electronic configuration in other carbon-based compounds, such as $Li_2CO_3$, $RO(CO_3)Li_2$, PAA, etc. Thus, CA-binder regions could be distinguished from other non-conductive compounds via mapping, so that the conductive composition distribution (i.e. the conductive network) can be visualized. Next, we quantitatively determine the degree of similarity between signals collected from a selected domain

of SEI on a cycled $SiO_x$ (Fig. 4g) with the reference. Consequently, an EELS mapping (Fig. 4h) of the relative conductive composition concentration can be obtained (see detailed calculation descriptions in Supplementary Fig. 13). There is a continuous path (presented by red color) in the selected box, indicating an interconnected conductive network. This structure not only explains how $SiO_x$ particles remain electrochemically active in the presence of a thick SEI layer, but more importantly, also illustrates the role of SEI aging in the capacity decay. In our proposed model (Fig. 4i), some CA (e.g. carbon black) will merge into the SEI upon cycling, forming a connected conductive network for electrons. However, the unrestricted volume expansion/contraction results in excessive thickening of the SEI, which effectively dilutes the spatial concentration of conductive agents in the SEI and eventually breaks down the percolation network for electrons. Essentially, the aging of SEI on $SiO_x$ signifies a larger resistivity due to the lack of conductive passages in an enlarged space.

## Regulating SEI growth for enhancing electrode stability

Based on previous observations, the free growth of the SEI layer is identified as a main cause for SEI aging (Fig. 5a). If one can effectively regulate the SEI thickening process by confining the unlimited expansion of $SiO_x$, the capacity fading might be mitigated. Recently, Meng and co-workers demonstrated that micro-sized Si can be stably cycled via applying a large external force[30]. Herein, a more practical

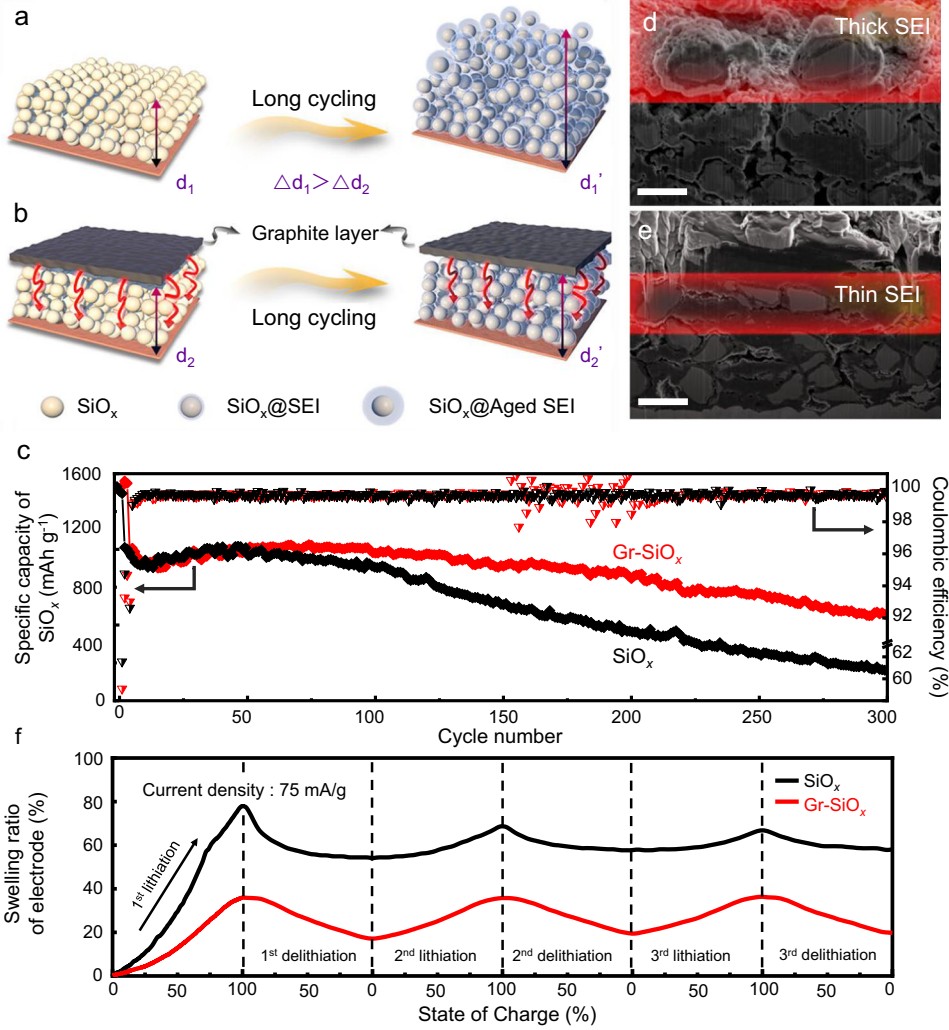

**Fig. 5 | Volume confining solution to improve electrode stability. a, b** Proposed schematic illustration of electrode evolution without and with a confining layer, respectively. **c** Comparison of cycling performance between cells with and without the graphite confining layer. **d** Cross-sectional SEM images of $SiO_x$ electrodes without graphite confining layer, where a thick SEI can be observed. **e** Cross-sectional SEM images of $SiO_x$ electrodes with a graphite confining layer, where thinner SEI is grown on $SiO_x$ particles. Scale bars in d and e = 5 um. **f** Swelling ratios of $SiO_x$ and $Gr-SiO_x$ electrodes during the initial 3 cycles.

strategy is proposed by coating an additional graphite confining layer on top of the $SiO_x$ electrode, applying a longitudinal force to mechanically restrict the SEI growth and the overall deformation of the electrode (Fig. 5b).

The effect of the confining layer was evaluated in half-cell test and the result is compared to the cell without a confining layer (Fig. 5c). Within the initial 50 cycles, the equivalent specific capacity curves of $SiO_x$ in the two cells almost overlap. After 50 cycles, $SiO_x$ with graphite confining layer ($Gr-SiO_x$) exhibited much slower capacity degradation. After 300 cycles, the equivalent specific capacity of the $Gr-SiO_x$ remained at 849 mAh $g^{-1}$, which is much higher than that of the pristine $SiO_x$ electrode (556 mAh $g^{-1}$). To better understand the effect of the graphite confining layer on the improved cycling performance, cross-sectional images of both electrodes after 300 cycles were examined (Fig. 5d and e). For the electrode without a graphite confining layer, a thick SEI layer of the topmost $SiO_x$ particles can be observed around the $SiO_x$ particles. By sharp contrast, a much thinner SEI was formed in the presence of a graphite confining layer. To better understand the working mechanism of the graphite confining layer, the swelling ratios of two electrodes are monitored during different state of charges (Fig. 5f). It is clearly shown that upon the initial lithiation, the $Gr-SiO_x$ electrode exhibits a much lower swelling ratio (36.0%) compared with pristine $SiO_x$ electrode (78.1%) and this discrepancy cannot be

recovered during subsequent cycles. Also, it should be noted that after experiencing the drastic volume expansion-contraction during the 1st cycle, the structure of the $SiO_x$ electrode became loose, creating more rooms for free particle expansion/contraction during subsequent cycles, hence the smaller thickness variation. Since the intrinsic volume changes of bulk $SiO_x$ can be hardly confined by such a layer, it can be concluded that the coating layer has suppressed the electrode expansion by exerting pressures to ensure the close packing of $SiO_x$ particles. As a result, the free space (i.e., voids) available for SEI growth can be significantly compressed. Additionally, the graphite layer remained on the top after 300 cycles (Supplementary Fig. 14). Considering this strategy can be easily integrated in a scale-up manufacturing of $SiO_x$ electrodes at low costs, it is worthy of further studies in the future.

To sum up, through combining of advanced characterization techniques, we dissected the morphological evolution of SEI on $SiO_x$ during long cycling. There are three key findings from this work. First, a 3D SEM image of SEI is reconstructed by sequential slicing of the electrode, where two types of SEI structures can be observed on the topmost $SiO_x$ particles, which follow an incrassation-densification evolution model. Second, by directly measuring the conductivity of SEI via nano-probing and visualizing the conductive network within the SEI via EELS mapping, we revealed the electron transfer through the percolation network of conductive agents within the thick SEI. Third,

based on the above findings, we conclude that the free growth of SEI accelerates its aging process. Therefore, a proof-of-concept approach is demonstrated by applying a graphite confining layer on top of the $SiO_x$ electrode, which greatly improved the cycling stability of the $SiO_x$ electrode by compressing the free space for SEI growth. The concept presented in this study not only unveils the aging mechanism of SEI evolution in $SiO_x$, but also defines an exciting way to fully tap the potential of $SiO_x$ anodes.

## Methods

### Preparation of electrodes and cell assembly

For the $SiO_x$ electrodes, a slurry was first prepared by dispersing $SiO_x$ microparticles (SL650A-SOC by Tianmu Energy Anode Materials Co., Ltd.), polyacrylic acid binder (3 wt% aqueous solution) and acetylene black in water with a weight ratio of 6:2:2. The slurry was cast onto a copper (Cu) foil, dried at room temperature for 4 h and further dried at 80 °C overnight under vacuum. The resulted $SiO_x$ electrodes (diameter = 10 mm) shows a mass loading of $1.2 \pm 0.1 \, mg \, cm^{-2}$, corresponding to an areal capacity of $1.8 \pm 0.18 \, mAh \, cm^{-2}$. The $SiO_x$ electrode processing is the same as that of commercial graphite electrodes without any additional pretreatment or prelithiation. For the $SiO_x$ electrode with a graphite confining layer, a similar protocol was applied. The weight ratio of graphite slurry was changed to 8:1:1, and the areal capacity of the graphite layer is no more than $0.1 \, mAh \, cm^{-2}$. The porosity of pristine $SiO_x$ and graphite coated electrodes is measured to be 17.5% and 20.3%, respectively using $N_2$ – adsorption method (Ultra-pycnometer, Quantachrome). CR2032 coin-type half-cells were assembled by sandwiching one piece of polyethylene separator (diameter = 20 mm, Celgard) between the $SiO_x$ electrodes and Li metal foil (diameter = 15.6 mm, thickness = 1 mm). The electrolyte (60 uL) used for cell assembly was $1.0 \, M \, LiPF_6$ in 90 vol % 1:1 (w/w) EC/DMC with 10 vol % fluoroethylene carbonate (FEC).

### Electrochemical measurements

Coin cells were tested in a battery testing system (Landt Instruments) and cycled between 0.05 and 1 V versus $Li/Li^+$ at $750 \, mA \, g^{-1}$ (to fully activate the $SiO_x$ electrode, the first three cycles were cycled between 0.05 and 1 V at a rate of $75 \, mA \, g^{-1}$). Charge/discharge rates were calculated assuming the theoretical capacity of $SiO_x$ is $1500 \, mAh \, g^{-1}$. Coulombic efficiency was calculated using the ratio of delithiation ($C_{dealloy}$) capacity to lithiation ($C_{alloy}$) capacity ($C_{dealloy}/C_{alloy} \times 100\%$). All electrochemical measurements were carried out at 25 °C.

### Electronical conductivity measurements

The electronic conductivities of the micro-composites were measured by an Imina Technologies sProber and Keithley 4200-CVU unit, and the radius of the probe tip was 500 nm. This nanoprobing and electronical conductivity analysis were performed in a Zeiss Sigma FEG SEM, and super flat disk-type specimens need to be prepared before the measurement. The $SiO_x$ electrodes were firstly washed in dimethyl carbonate (DMC), and embedded in a soft and commercial epoxy resin (EpoFix, Struers Co., Ltd.). After curing the resin, the samples were sequentially polished with 1200-4000 mesh silicon carbide sand paper. Final polishing was made by a Leica EM TIC 3X Ar ion beam milling system: 20 min cleaning at 5 kV and 15 min polishing at 1 kV. The above experiment was carried out at room temperature.

### Optical microscopy

The super flat disk-type specimens mentioned above were investigated by Keyence VK-X1100 3D Laser scanning microscope at a magnification of 1000X. The above experiment was carried out at 25 °C.

### X-ray photoelectron spectroscopy (XPS) measurement

The XPS (ESCALAB 250XL) test was used to analyze the element distribution on the electrode surface at different charging/discharging states. All the samples have been transferred in vacuum and during XPS measurements, the base pressure of sample chamber was kept below $3.0 \times 10^{-10}$ mbar. The above experiment was carried out at 25 °C.

### 3D FIB-SEM tomographic imaging

An integrated FIB-SEM system consisting of FEI SCIOS (FIB) and ZEISS SUPRA® 55 (2D SEM) is employed for tomographic imaging. The electrodes were firstly washed with DMC to remove any residual Li salts from the electrode surface and then transferred from the glove box to the FIB-SEM using a transfer vessel to avoid any exposure to air. Areas for observation were precoated with ~1 μm of platinum (Pt) via a Pt gun based on thermal evaporation in the FIB-SEM chamber. FIB trenches through the indentation sites were sputtered using 30 kV $Ga^+$ ions and beam currents of 0.5–3 nA. For 3D FIB tomography, the gallium focused ion beam was used at normal incidence to locally cross-section the indentation sites, and SEM secondary electron (SE) images were taken of each sequential x-y 2D slice at a tilt angle of 52°. The dimensions of the FIB cutting area for different $SiO_x$ particles were $x = 25 \, μm$, $y = 25 \, μm$, $z = 12-15 \, μm$. A magnification of 5000–10,000 times was used, and the spacing, z, between adjacent 2D FIB sections was 50 nm. The scanning zone of SEM was aligned and corrected automatically during FIB cutting process, by using Auto Slice software to recognize the cross-correlation of features on the surfaces of the sample. The 3D particle models in Fig. 2f-j were reconstructed by using the Avizo software. The above experiment was carried out at 25 °C.

### Electron Microscopy

To prepare the TEM specimen, a layer of platinum (~500 nm thickness) was deposited to protect the first atomic layers on the surface of the $SiO_x$ particle during the FIB cut (30 kV, 0.5-3 nA). Because of the high sensitivity of lithiated materials, the electron gun was preferred to the ion gun to carry out the deposition. To minimize amorphization and damages caused to the cross section, an ionic polishing step was carried out using an Ar-ion polishing system (Gatan Precision Ion Polishing System, PIPS II Model 695) under 2 keV/30 μA. All sample preparation processes were performed in vacuum/Ar atmosphere. The sample transfer from FIB to TEM was carried out in a custom-made Ar filled glove box. All conventional STEM characterizations were carried out using an FEI Titan 60-300 (scanning) transmission electron microscope operated at an accelerating voltage of 300 kV with an energy resolution of 0.8 eV. The instrument is equipped with an aberration corrector in the image-forming lens, which was tuned before each sample analysis. The beam was blanked between images to minimize total electron dose. STEM-EELS characterization was performed with a C2 aperture of 50 μm, a beam current of ~100 pA, a camera length of 29.5 mm, and a pixel dwell time of 0.1 s. These settings give a convergence angle of 25.0 mrad and an acceptor angle of 56.5 mrad. EELS spectra were acquired on a GIF Quantum with a dispersion of 0.1 eV/channel in Dual EELS mode, with the low-loss centered on the zero-loss peak and the core-loss centered on the C K-edge. Energy drift during spectrum imaging was corrected by centering the zero-loss peak to 0 eV at each pixel. The EELS spectra and the spectrum image were summed over many pixels on the $SiO_x$/SEI domains and smoothed with a Savitsky-Golay filter. The above experiment was carried out at 25 °C.

### Electrode Swelling tests

$SiO_x$ and $Gr-SiO_x$ electrodes are coupled with $LiCoO_2$ electrode (with N/P ratio of 1.07) in a home-made Swagelok cell. The mass loading of $SiO_x$ and $LiCoO_2$ are $1.2 \pm 0.1 \, mg \, cm^{-2}$ and $9.6 \pm 0.1 \, mg \, cm^{-2}$, respectively. The cells are tested in in a battery testing system (SCT-S, Arbin) and cycled between 3.0 and 4.2 V at $75 \, mA \, g^{-1}$. The swelling ratio ($\Delta d/d_0$, where $\Delta d$ is the thickness variation of the electrode and $d_0$ is the initial thickness of the electrode) is monitored using a displacement sensor. The above experiment was carried out at 25 °C.

## Elemental analysis

Different SiO$_x$ samples were dissolved in HF solution, then diluted for inductively coupled plasma optical emission spectrometry (ICP-OES) measurement (Agilent ICP-OES 725ES). The above experiment was carried out at 25 °C.

## Data availability

Source data are provided with this paper. Additional data supporting the findings of this study are available from the corresponding author upon request. Source data are provided with this paper.

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

## Acknowledgements

G.Q. acknowledges financial support from Young Elite Scientists Sponsorship Program by CAST (No. YESS20220618), Shenzhen Science and Technology Program (Grant No. RCBS20200714114820077), and Hundreds of Talents program of Sun Yat-sen University (No. 210192). L.Y. and F.P. acknowledges financial support from Shenzhen Science and Technology Research Grant (ZDSYS201707281026184) and Shenzhen Science and Technology Planning Project (JSGG20220831095604008). X.L. acknowledges financial support from Soft Science Research Project of Guangdong Province (No. 2017B030301013), and Guangdong Basic and Applied Basic Research Foundation (2021B1515120002). K.A. and T.L. acknowledge financial support from Clean Vehicles, US-China Clean Energy Research Centre (CERC-CVC2) under the US DOE EERE Vehicle Technologies Office. Argonne National Laboratory is operated for the DOE Office of Science by the UChicago Argonne, LLC, under contract no. DE-AC02-06CH11357.

## Author contributions

G.Q. and Y.L. contributed equally to this work. G.Q., Y.S., L.Y. and F.P. conceived and designed the experiments. F.P. directed the project. Y.L., H.C., T.L., Y.S. and X.Y. prepared the materials and performed the electrochemical testing. N.Y., H.Lin and X.L. conducted the FIB-SEM characterization. L.X., C.L., J.C., N.N. and W. Z. performed the electron microscopy and STEM-EELS measurements. M.Z. and Z.L. carried out the electrode expansion experiment. G.Q., C.L. and L.Y. analyzed the data. G.Q., H.C and L.Y. wrote the paper. L.Y., H.Li, K.A., L.C. and F.P. provided materials and characterization tools. K.A. and F.P. polished the writing. All authors discussed the results and commented on the manuscript.

## Competing interests

The authors declare no competing interests.
