## [Peer Review file · Nature Communications]

Reviewers' comments:

Reviewer #1 (Remarks to the Author):

Title: Revealing the Aging Process of Solid Electrolyte Interphase on SiO_x Anode

In this paper, the authors first revealed an exceptionally characteristic SEI microstructure with a nanoporous inner region and a dense outer region, then discovered that the electronic conductivity of thick SEI relies on the percolation network within composed of conductive agents. Finally, the authors adopted a proof-of-concept strategy to mechanically restrict the SEI growth via applying a graphite layer on top of the electrode. I recommend the acceptance of this manuscript after the following issues are well addressed

1. In introduction, the authors mentioned “to reveal the forming process of SEI on SiO_x materials, new approaches should be adopted”, but they didn’t indicate in what specific ways the existing testing techniques do not meet the needs of the current work.
2. The authors are suggested to pay more attention to check the full manuscript, since some errors are present. For example, the authors mentioned Fig. 2j k in the manuscript, but there is no corresponding Fig. 2k in Fig. 2 at all. In addition, the description of Fig. S8 is divided into Fig. S8a,b,d and d, but Fig. 8 is not marked with a-d diagram. G
3. In page 6, the authors mentioned that “After 50 cycles, the thickness of SEI layer increased from 100 nm to approximately 250 nm”, but no description of “100 nm” could be found in the previous paragraphs.
4. The authors believed that the conductivity of SEI comes from the carbon black particles doped by outward expansion during SEI aging, but why the trend of SEI resistance measured in Fig. 4d is consistent with SiO_x.
5. The authors are suggested to confirm whether Fig. S7d is the enlarged area corresponding to Fig. S7c, and if so, the corresponding area should be marked in Fig. S7c.
6. The authors declared that the aging and thickening process of SEI gradually increases from inside to outside, and gradually densifies, then should the outermost SEI be the most stable? However, in the description of Fig. S9, the authors declared that "SEI is not completely stable even after a long cycle". Is this contradictory?
7. The authors declared that “After the first lithiation, a porous SEI layer could be observed around the particle. Surprisingly, this SEI layer dramatically diminished after the first delithiation, which is supported by the XPS results (Supplementary Fig. 8).” I want to know what’s the reason for no C1s peak could be found in the 1st lithiation, 300th lithiation/delithiation.

Reviewer #2 (Remarks to the Author):

Qian et al. performed a very detailed study of the microstructural behavior of a SiO_x negative electrode material through very high level characterization techniques. A considerable and delicate work was performed on a very important matter. In particular FIB sectioning-3D reconstruction as well as thin lamella for TEM and STEM-EELS were employed to give an image of the SEI at a nanometer level. Some of the experiments are very innovative (such the conductivity measurement at the nanoscale) and deserve high attention. They developed an interesting concept of type I and type II SEI.

Unfortunately, some of these experiments, decisive to the study of the authors, did not convince me in relation to preservation of the real sample supporting the conclusions. Considering the work to be further performed to accomplish that, I do not support publication in Nature communications.

Indeed, in the case of the characterization of SEIs, due to its particularly sensitivity to interacting beams (photons, electrons, ions...), a special attention must be paid to the dose and the experimental conditions. I do not find enough of such concern in the work and I give many examples in the following to justify my opinion.

In Fig2, a "type I SEI" is proposed. But Ga⁺ ion, because of particular milling condition of porous (inhomogeneous) materials often amorphize/make porous surface of samples. I have seen it on samples not even containing Silicon. Based on this, I find the Fig3 a bit overstating 2 types of SEI.

In XPS spectra, the Li peak would be worth showing. But on these composite electrodes, one can always wonder what is the use of XPS when the beam is roughly 300 μm x 500 μm. Hence, the variations observed in Supp Fig9 are rather uncertain.

XPS without argon clusters (Ar⁺ used in the study) would probably damage a lot the SEI and make the results mostly irrelevant to the real chemical state of the surface.

Polishing with ion-beam milling usually provokes sensible local increase of temperature so no idea if the contrast seen on some images is real (next to SiO_x particles). (for example Supplementary Fig 6).

The TEM lamella, probably already modified by the FIB preparation, is surely very beam sensitive and I doubt high losses spectra can be obtained without damaging the sample containing Li₂O, LiF... It would be interesting to see an image of the area after the analysis... How was transferred the TEM lamella from the FIB to the TEM?

Other comments:

No comparison with other formulation of electrodes demonstrating a worse cycling when less CA in the binder is used. The quantity of it is quite high in mass, so volume wise, there is basically acetylene black everywhere.

Washing electrodes with DMC is always a risky task: not reproducible, chemically selectiveness (soluble species) ...

The embedding in a resin is a typical preparation technique for various samples (Suppl Fig 10). However, these chemical compounds are not inert ones (by definition, they need to be polymerized).

Consequently, I am very doubtful on the preservation of the "real SEI" in these measurements (even when delithiated...).

In Suppl. Fig 12, the "SEI" part looks more like a hole due to beam damage than a solid (porous) phase. Hence, I do not find the electric conductivity measurement on this part convincing enough.

The various EELS spectra shown in Suppl. Fig 13 are diverse and a simple extraction of CA in them to get Fig 4h, is highly speculative, at least without further detail. Furthermore, Acetylene black is a CA with a very high-level sp² carbon and EELS K-edge spectra usually present a much sharper π^* peak, so I find that the results do not demonstrate the clear presence of the CA in the SEI. Maybe high resolution TEM (corrected) images could provide such information.

I find the "confining" idea of the SEI not clear at all, at least in its set-up and effectiveness: what about the pressure applied on the whole cell? What about the movement of the graphite "layer" along cycling (they could change quite substantially the electronic conductivity after moving inward within the electrode)?

Furthermore, apart from the first cycle, subsequent ones have a smaller swelling ratio variation in the case of pure SiO_x...

When swelling is concerned in electrodes, a parameter to control is usually the porosity of the electrode. Nothing is said about that in the manuscript.

Other remarks:

The 2kV beam current was surely not 180 μ A!

I do not see the 1 micro Pt coating for example in Fig 2. How was it deposited?

What is a compliant epoxy resin?

"FIB-SEM (FIE SCIOS-ZEISS Supra55)": both FIBs were used? For which experiments?

There is no Fig 2k.

The least to do is to give a value for x in SiO_x. One could wonder if similar results are obtained for different values of x and it is necessary to be able to reproduce exposed results.

Reviewer #1

General Comments: *In this paper, the authors first revealed an exceptionally characteristic SEI microstructure with a nanoporous inner region and a dense outer region, then discovered that the electronic conductivity of thick SEI relies on the percolation network within composed of conductive agents. Finally, the authors adopted a proof-of-concept strategy to mechanically restrict the SEI growth via applying a graphite layer on top of the electrode. I recommend the acceptance of this manuscript after the following issues are well addressed*

Response: We are truly thankful to the reviewer for the positive assessment of our work, and for providing valuable and insightful comments that helped us to reflect on the scientific significance of our study. We sincerely hope that this revision relieves the reviewer's concerns.

Comment #1: *In introduction, the authors mentioned "to reveal the forming process of SEI on SiOx materials, new approaches should be adopted", but they didn't indicate in what specific ways the existing testing techniques do not meet the needs of the current work.*

Response: We thank the reviewer for the constructive comment. Indeed, we should have specified the shortage of the existing characterization techniques. To better reflect the novelty and significance of our work, following revisions have been made to the manuscript:

Main manuscript:

However, these above-mentioned techniques are unsuitable for directly observation of the structural evolution of SEI on micro-sized particles with large volume swings, while traditional microscope-based technique is only suitable for 2D observation of the cross-sectional structure of the electrode without 3D information. In this case, new approaches should be adopted.

Comment #2: *The authors are suggested to pay more attention to check the full manuscript, since some errors are present. For example, the authors mentioned Fig. 2j k in the manuscript, but there is no corresponding Fig. 2k in Fig. 2 at all. In addition, the description of Fig. S8 is divided into Fig. S8a,b,d and d, but Fig. 8 is not marked with a-d diagram.*

Response: We apologize for these mistakes. After carefully checking all figure numbers, corrections have been made accordingly in the manuscript:

Main manuscript:

Surprisingly, this SEI layer dramatically diminished after the first delithiation, which is supported by the XPS results (Supplementary Fig. 7b and 7c).

A key finding from Fig. 2d and 2e is that, after long-term cycling...

Supplementary Information:

Supplementary Fig. 7 | Full XPS spectra of the SiO_x electrode a, before cycling; b, after 1st lithiation; c, 1st delithiation; d, 300th lithiation and e, 300th delithiation.

As shown in Supplementary Fig. 7a, C, Si, O and F element could be observed in SiO_x electrode before cycling. However, the peak related to Si disappeared and the peaks for O and F element became more distinct after the initial discharging, meaning that the formation of SEI on the surface. Interestingly, as shown in Supplementary Fig. 7b and 7c, a weak peak for Si element could be found in the specimen at the 1st delithiation, implying the SEI became thinner after the initial charging process. This result can be further verified by the weakened C-C peak (corresponding to conductive carbon) after 1st lithiation and the intensified C-C peak after 1st delithiation. The Si peak could not be found in the 300th delithiated and lithiated specimens, demonstrating the thick SEI on the surface of SiO_x anode after long-term cycles.

Comment #3: *In page 6, the authors mentioned that “After 50 cycles, the thickness of SEI layer increased from 100 nm to approximately 250 nm”, but no description of “100 nm” could be found in the previous paragraphs.*

Response: We apologize for the mistake and we thank the reviewer for the scrutinization. Previously, we tried to estimate the thickness of SEI during the early stage of cycling. However, we find the variation too large to be statistically sound. Therefore, the description of “100 nm” should have been deleted with other relevant discussions. The sentence has been re-written accordingly:

Main manuscript:

After 50 cycles, the thickness of SEI layer **has grown** to approximately 250 nm.

Comment #4: *The authors believed that the conductivity of SEI comes from the carbon black particles doped by outward expansion during SEI aging, but why the trend of SEI resistance measured in Fig. 4d is consistent with SiO_x.*

Response: We thank the reviewer for the insightful comment. Actually, if we look at the zoomed-in curve of CA-binder domain (**Fig. R1b**), it shows a similar trend with SiO_x and SEI region near zero bias (from -0.1 V to 0.1 V). This trend originates from the Schottky barrier between the metal detector and semi-conductors, the barrier will be overcome at high voltage. However, for CA region, as the voltage increases over 0.15 V, the interface can be seen as Ohm contact. In this case, the charge carriers of components in CA-binder domain are saturated, hence a flat I-V curve (**Fig. R1c**). The reason why SEI resembles to SiO_x is because despite containing some conductive carbon, the over-all features of SEI resemble semiconductors.

Fig. R1 | a, Resistivity vs. voltage curves in different regions. b, Zoomed-in curve of CA area near zero bias; c, I-V curve of the CA area.

Comment #5: *The authors are suggested to confirm whether Fig. S7d is the enlarged area corresponding to Fig. S7c, and if so, the corresponding area should be marked in Fig. S7c.*

Response: We thank the reviewer for the useful comment. We have chosen a different zoom-in image, which is included in Supplementary Fig. 6c in the revised version:

Supplementary Information:

Supplementary Fig. 6 | The cross-sectional SEM images of the SiO_x electrode (a-b) before cycling, and (c-d) after 1st delithiation in different magnification.

Comment #6: *The authors declared that the aging and thickening process of SEI gradually increases from inside to outside, and gradually densifies, then should the outermost SEI be the most stable? However, in the description of Fig. S9, the authors declared that "SEI is not completely stable even after a long cycle". Is this contradictory?*

Response: We thank the reviewer for the insightful comments. The polymeric compositions in the SEI may be re-oxidized during delithiation, hence the change in the surficial chemical composition, which is reflected in the XPS results. We apologize for the misleading description. After careful deliberation, we have decided to drop Supplementary Fig. 9 as the chemical composition of SEI is not important in this study and this figure does not help us to better demonstrate our model.

Comment #7: *The authors declared that "After the first lithiation, a porous SEI layer could be observed around the particle. Surprisingly, this SEI layer dramatically diminished after the first delithiation, which is supported by the XPS results (Supplementary Fig. 8)." I want to know what's the reason for no C1s peak could be found in the 1st lithiation, 300th lithiation/delithiation.*

Response: We really appreciate the inspiring comments from the reviewer. Please note that every electrode in this figure shows C 1s peak, only with difference in their peak intensity. We believe the relatively strong C 1s peak shown in the pristine electrode belongs to conductive carbon. The 1st lithiated electrode is covered with polymeric decomposition products, hence the weakened C 1s peak. Conductive carbon re-exposed as the previously formed SEI disintegrated during delithiation, which supports our model demonstrated in Fig. 3a:

After 300 cycles, the SEI became too thick to be easily destroyed, hence the C 1s peak will not be recovered after delithiation, which supports our model demonstrated in Fig. 3c.

Again, we would like to thank the reviewer as this insightful comment has inspired us to better demonstrate our claims. Relevant discussions have been added accordingly:

Main manuscript:

An SEI layer is initially formed on the expanded SiO_x upon the first lithiation, which is supported by the diminished C 1s and Si 2p XPS signal (Supplementary Fig. 7). During the first delithiation, the freshly formed Type I SEI cannot shrink at the same pace as the particle due to the lack of adhesion force between them, leading to partial detachment from SiO_x particles (Fig. 3a), which is supported by the reappeared C 1s and Si 2p peaks (Supplementary Fig. 7) as well as the gradual increase of coulombic efficiency during initial 5 cycles (Supplementary Fig. 11).

Supplementary Information:

Interestingly, as shown in Fig. 7b and 7c, a weak peak for Si element could be found in the specimen at the 1st delithiation, implying the SEI became thinner after the initial charging process. This result can be further verified by the weakened C-C peak (corresponding to conductive carbon) after 1st lithiation and the intensified C-C peak after 1st delithiation. The Si peak could not be found in the 300th delithiated and lithiated specimens, demonstrating the thick SEI on the surface of SiO_x anode after long-term cycles.

Reviewer #2

General Comments: *Qian et al. performed a very detailed study of the microstructural behavior of a SiO_x negative electrode material through very high level characterization techniques. A considerable and delicate work was performed on a very important matter. In particular FIB sectioning-3D reconstruction as well as thin lamella for TEM and STEM-EELS were employed to give an image of the SEI at a nanometer level. Some of the experiments are very innovative (such the conductivity measurement at the nanoscale) and deserve high attention. They developed an interesting concept of type I and type II SEI. Unfortunately, some of these experiments, decisive to the study of the authors, did not convince me in relation to preservation of the real sample supporting the conclusions. Considering the work to be further performed to accomplish that, I do not support publication in Nature communications. Indeed, in the case of the characterization of SEIs, due to its particularly sensitivity to interacting beams (photons, electrons, ions...), a special attention must be paid to the dose and the experimental conditions. I do not find enough of such concern in the work and I give many examples in the following to justify my opinion.*

Response: We sincerely appreciate the time and effort taken in reviewing our manuscript and the constructive comments provided. We tried our best to address the reviewer's concerns about the sample preservation during preparation and characterization, and carefully revised the manuscript accordingly. Also, we hope the reviewer to kindly consider that the observation of SEI on SiO_x might be different to other anode materials (e.g. graphite): its much thicker (under submicron scale) and resilient against beam damage. We are truly thankful for the constructive comments that would significantly improve this work and we also hope that this revision relieves the reviewer's concerns.

Comment #1: *In Fig2, a "type I SEI" is proposed. But Ga⁺ ion, because of particular milling condition of porous (inhomogeneous) materials often amorphize/make porous surface of samples. I have seen it on samples not even containing Silicon. Based on this, I find the Fig3 a bit overstating 2 types of SEI.*

Response: We thank the reviewer for the insightful comments. We do understand your concerns regarding to the beam damage to the sample. Admittedly, we cannot guarantee perfect sample preservation since FIB is indeed a constructive sample preparation method. However, whether the beam damage will interfere our observation depends on how the sensitivity of sample to beam and

the scale of observation. In order to verify the suitability of our preparation method, we have also prepared sample through cryo-ultramicrotomy, where samples were sliced up mechanically under cryogenic environment. This is one of the best methods we can think of to reveal the cross-sectional structure of SiO_x anodes with minimum sample damage. It should be noted that being unable to connected with an imaging system (e.g. SEM), cryo-ultramicrotome was not adopted in this work since we aim to obtain a 3D reconstructed graph. As a result, very similar SEI structures can be observed. From this comparison, we can conclude that beam damage will not affect SEI morphology at sub-micron scales.

Manuscript:

Additionally, to preclude the interference of beam damage to the sample, cryo-ultramicrotomy was also carried out to prepare the sample. As shown in Supplementary Fig. 10, both methods result in similar SEI morphology, suggesting beam damage is negligible in this study.

Supplementary Information:

Supplementary Fig. 10 | Cross-sectional SEM images of SiO_x anode after 300 cycles prepared from a,b, cryo-ultramicrotomy and c,d, FIB.

Regarding to the reviewer's concern that we might have overstated two types of SEI, we would like to point out that we did not mean to define them as two different types of SEI layer. An early-stage SEI (Type I) generally show a porous structure, during repeated cycling, it will develop into a dense SEI (Type II). We agree that there should not be a clear line to draw them apart as they might be similar in terms of chemical compositions. We used such terminology only to emphasize the process of SEI densification, to avoid unnecessary confusion, additional discussions are added to the revised manuscript:

Manuscript:

Note that Type I and Type II SEI only differ in the morphology while they might exhibit very similar chemical compositions.

Comment #2: *In XPS spectra, the Li peak would be worth showing. But on these composite electrodes, one can always wonder what is the use of XPS when the beam is roughly 300 μm \times 500 μm . Hence, the variations observed in Supp Fig9 are rather uncertain.*

Response: We thank the reviewer for the comments. We agree that the small variation in XPS might not be statically significant. Considering that the chemical composition of SEI is quite irrelevant and the information does not help us to better demonstrate the main claims of this work, we have decided to drop Supplementary Fig. 9 from this work to void unnecessary distraction.

Comment #3: *XPS without argon clusters (Ar^+ used in the study) would probably damage a lot the SEI and make the results mostly irrelevant to the real chemical state of the surface.*

Response: We thank the reviewer for the comments. We agree that Ar^+ etching could potentially change the chemical state of SEI. However, if we are just looking at the full spectra, we can see that Ar^+ etching barely makes any difference as the etching process will only cause slight chemical shift.

In this work, Ar^+ was adopted because a thick SEI on the very top of the electrode (not necessarily on SiO_x) has prevented us to observe the SEI on SiO_x . For instance, the difference of Si signal between 1st lithiation and delithiation becomes more distinctive without Ar^+ etching:

Fig. R3 | Comparison between XPS results of the SiO_x anode a-e, after and f-j, before Ar⁺ etching.

Comment #4: *Polishing with ion-beam milling usually provokes sensible local increase of temperature so no idea if the contrast seen on some images is real (next to SiO_x particles). (for example Supplementary Fig 6).*

Response: We thank the reviewer for the comments. Indeed, after reconsideration, we find the conclusion drawn from Supplementary Fig. 6 in the original manuscript, which is an optical photo, might be far-fetched. Therefore, we have dropped this figure and deleted relevant discussions:

~~The corresponding optical images are shown in Supplementary Fig. 6. It can be observed that the pristine particles are covered by a matrix of carbon black and polymer binder, which serves as the electron percolation network of active materials.~~

As we have previously demonstrated in the response to Comment #1, FIB and ion-milling will not affect the structure of SEI in SEM images. 3X Ar-ion beam milling will cause the sample temperature to increase to approximately 70 °C, which should not cause significant damage to the main components of SEI. Moreover, even if ion-milling somehow caused the discoloration of SEI under optical microscope, it would not affect any conclusion we have drawn. Instead, such contrast

helps us to confirm that the region around SiO_x is indeed SEI by cross-referencing the SEM and the optical images (Supplementary Fig. 8d and 8e in the revised manuscript).

Comment #5: *The TEM lamella, probably already modified by the FIB preparation, is surely very beam sensitive and I doubt high losses spectra can be obtained without damaging the sample containing Li_2O , LiF ... It would be interesting to see an image of the area after the analysis... How was transferred the TEM lamella from the FIB to the TEM?*

Response: We thank the reviewer for the detailed comments. We understand the concerns from the reviewer, but these factors will not affect our conclusion for two reasons: (1) Map scanning mode was adopted in the EELS experiment, which could minimize the damage on the sample. (2) In this case, we used EELS to characterize the distribution of conductive carbon, which is far more stable against the beam compared with Li_2O and LiF , hence the conclusion of our work shall not be affected. Indeed, if we were studying the distribution of other beam-sensitive components, cryo-TEM with lower voltage (e.g. 80 kV) might be required.

The sample was transferred in the custom-made glove box as shown below, which avoids the contamination of moist and oxygen.

Fig. R3 | Pictures of sample transfer from FIB-SEM to HRTEM.

Relevant details have been added to the experimental section:

Manuscript:

The sample transfer from FIB to TEM was carried out in a custom-made argon-filled glove box.

Other comments:

Comment #6: *No comparison with other formulation of electrodes demonstrating a worse cycling when less CA in the binder is used. The quantity of it is quite high in mass, so volume wise, there is basically acetylene black everywhere.*

Response: We thank the reviewer for the insightful comments. Unlike commercially used ones, the SiO_x we used in this work was not carbon-coated. Therefore, as we used a different ratio of CA (10%), the electrode could not be properly discharged or charged. We believe the total amount of carbon will not necessarily affect our conclusion as we only focus on the carbon adjacent to SiO_x particles.

Fig. R4 | Cycling performance of a SiO_x anode with a higher (80 wt%) active material content.

Also, this result actually agrees with our conclusion: since there are less conductive agents around, the dilution of conductive network by SEI growth becomes more severe.

Comment #7: *Washing electrodes with DMC is always a risky task: not reproducible, chemically selectiveness (soluble species) ...*

Response: Despite this method is a very common way to pre-treat electrode before some specific characterizations, we agree that DMC might potentially wash away some soluble species on the surface, causing slight deviation of chemical components on the surface. We hope there is better way to obtain a clean electrode without any repercussion. Nevertheless, since we have excluded detailed XPS data from the manuscript, we believe such pre-treatment will not affect the main argument considering that we focus on sub-micro scale morphologies of SEI.

Comment #8: *The embedding in a resin is a typical preparation technique for various samples (Suppl Fig 10). However, these chemical compounds are not inert ones (by definition, they need to be polymerized). Consequently, I am very doubtful on the preservation of the “real SEI” in these measurements (even when delithiated...).*

Response: We thank the reviewer for the insightful comment. First, we are fully aware of the potential contamination by the resin. Therefore, we **only** used resin for the Nanoprobng experiment in Fig. 4a-d. Moreover, we used EpoFix epoxy resin mixed with EpoFix Hardener produced by Struers Co., Ltd. in this experiment (the detail has been added to the manuscript). This resin was deliberately chosen due to its high viscosity, which prevented it from penetrating the dense shell of SEI. In this case the conductivity measurement of the SEI should not be an issue by the resin embedding.

Manuscript:

...and embedded in a soft and commercial epoxy resin (EpoFix, Struers Co., Ltd.).

Comment #9: *In Suppl. Fig 12, the “SEI” part looks more like a hole due to beam damage than a solid (porous) phase. Hence, I do not find the electric conductivity measurement on this part convincing enough.*

Response: Thank you for your comments. Indeed, the area you mentioned seems slightly dented due to the brightness and contrast. Therefore, a figure with adjusted contrast was demonstrated below for better observation. By looking closely, it could be found that the whole area rather flat, and definitely not a hole. We would also like to point out that the electronic conductivity was measured by placing two probes on the surface of selected area. If it is a hole, there would be an open circuit and we will not be able to obtain any values. Therefore, we are confident about the area we measured is SEI.

Fig. R5 | Comparison between the SEM image in Supplementary Fig. 12 (origin submission) before (left) and after adjusting brightness and contrast (right).

Comment #10: *The various EELS spectra shown in Suppl. Fig 13 are diverse and a simple extraction of CA in them to get Fig4h, is highly speculative, at least without further detail. Furthermore, Acetylene black is a CA with a very high-level sp² carbon and EELS K-edge spectra usually present a much sharper π^* peak, so I find that the results do not demonstrate the clear presence of the CA in the SEI. Maybe high resolution TEM (corrected) images could provide such information.*

Response: We thank the reviewer for the insightful comments. We apologize for not providing details of how we extracted CA from the EELS spectra. First, we normalized the curve based on the peak between the region of 380-420 eV, which is away from the near-edge peaks of carbon.

Next, we used Point 0 in CA-binder domain as the reference to evaluate the resemblance between a certain point with Point 0 at the region of near-edge fine structure of carbon (280-295 eV) based on mean square error method shown as below:

Fig. R6 | Examples of how the mean square error was calculated. y_i and x_i are normalized value of the sample to be measured and Point 0, respectively.

A smaller M value represents a higher resemblance between this point, hence the chemical composition of this point in SEI is closer to the CA-binder domain. To avoid confusion, we have replaced all CA with CA-binder in the revised manuscript and calculation details are also added to the revised supplementary information:

Supplementary Information:

To determine the distribution of CA-binder, we normalized the curve based on the peak between the region of 380-420 eV, which is away from the near-edge peaks of carbon. Next, we used Point 0 in CA-binder domain as the reference (x_i) to evaluate the resemblance between a certain point (y_i) with Point 0 at the region of near-edge fine structure of carbon (280-295 eV) based on mean square error method shown as below:

$$M = \frac{\sum_{i=1}^n (y_i - x_i)^2}{N}$$

A smaller M value represents a higher resemblance between this point, hence the chemical composition of this point in SEI is closer to the CA-binder domain.

As we mentioned above, instead of existing alone, carbon black is always covered with polymeric binders, which will inevitably flatten the sharp π^* peak of carbon black. Apart from that, it has been pointed out by Papworth et al., that sp² carbon does not necessarily show a sharp π^* peak, which might be due to indirect transition from 1s to π^* state [Physical review B, 62.19 (2000):12628]. Also, it should be noted that the signal might be affected by the adjacent polymeric binder, which will further weaken the π^* peak.

Comment #11: I find the “confining” idea of the SEI not clear at all, at least in its set-up and effectiveness: what about the pressure applied on the whole cell? What about the movement of the graphite “layer” along cycling (they could change quite substantially the electronic conductivity after moving inward within the electrode)? Furthermore, apart from the first cycle, subsequent ones have a smaller swelling ratio variation in the case of pure SiOx...

Response: It is a very good question and we should have explained this better in the manuscript. First, we would like to point out that the pressure applied on coin-cells cannot directly apply to the particles, but only to the cell shells to make sure good sealing and interfacial contact. Also, the small “spring” in coin cells can only provide a very, very limited kickback force upon electrode volume expansion. Therefore, the pressure applied on the cell could barely confine particle volume expansion. The role of the graphite coating layer is more like an electrochemically active artificial protective layer to regulate (not fully suppress) the overall electrode expansion.

Next, from the SEM image shown below, it can be observed that the SiO_x particles remain in the lower part of the electrode after 300 cycles. Therefore, the movement of graphite into the electrode will not be an issue since the expansion and contraction of SiO_x only lead to vertical movement of graphite layer, and there is no apparent horizontal stress and deformation to cause the breakdown of the top layer.

As for the last question, please note that the swelling ratio is measured based on the thickness variation of the electrode. After experiencing the first large volume expansion-contraction, the structure of SiO_x electrode became loose after the 1st cycle, creating more rooms for free particle expansion/contraction during subsequent cycles, hence the smaller thickness variation. To avoid confusion, discussions have been added to the manuscript accordingly:

Manuscript:

Also, it should be noted that after experiencing the drastic volume expansion-contraction during the 1st cycle, the structure of SiO_x electrode became loose, creating more rooms for free particle expansion/contraction during subsequent cycles, hence the smaller thickness variation. Since the intrinsic volume changes of bulk SiO_x can be hardly confined by such a layer, it can be concluded that the coating layer has suppressed the electrode expansion by exerting pressures to ensure the close packing of SiO_x particles. As a result, the free space (i.e., voids) available for SEI growth can be significantly compressed. Additionally, the graphite layer remained on the top after 300 cycles (Supplementary Fig. 14).

Supplementary Information:

Supplementary Fig. 14 | Cross-sectional SEI image of a graphite coated SiO_x anode after 300 cycles and the corresponding EDS mapping of Si. It can be seen that a graphite layer layer (~5 μm) is firmly covered on the top while Si remain in the lower part of the electrode.

Comment #12: *When swelling is concerned in electrodes, a parameter to control is usually the porosity of the electrode. Nothing is said about that in the manuscript.*

Response: Thank you for your constructive suggestions. We have measured the porosity of the electrode using N₂-adsorption method. It is shown that the graphite coated electrode shows slightly higher porosity (20.3%) compared to the pristine one (17.5%), which could be due to the small gap between the coating layer and SiO_x electrode. The corresponding results are added to the manuscript:

Manuscript:

The porosity of pristine SiO_x and graphite coated electrodes is measured to be 17.5% and 20.3%, respectively using N₂ - adsorption method (Ultra-pycnometer, Quantachrome).

Other remarks:

Comment #13: *The 2kV beam current was surely not 180 μA!*

Response: Thank you for the notice. Sorry for the mistake. Revisions have been made to the manuscript:

Manuscript:

To prepare the TEM specimen, a layer of platinum (~500 nm thickness) was deposited to protect the first atomic layers on the surface of the SiO_x particle during the FIB cut (30 kV, 0.5-3 nA). Because of the high sensitivity of lithiated materials, the electron gun was preferred to the ion gun to carry out the deposition. To minimize amorphization and damages caused to the cross section, an ionic polishing step was carried out by using an ion milling system (2 keV/30 μA). All sample preparation processes were...

Comment #14: *I do not see the 1 micro Pt coating for example in Fig 2. How was it deposited?*

Response: The coating can be observed in the cross-sectional image below. Pt was deposited on the electrode via a Pt gun in the FIB chamber:

Fig. R7 | a, Cross-sectional SEM image of a cycled SiO_x electrode after Pt coating. b, SEM image of a Pt gun above the substrate.

Relevant details have been added in the experimental section:

Manuscript:

The FIB-SEM samples were pre-coated with ~1 μm of platinum (Pt) via a Pt gun in the FIB-SEM chamber to prevent charging and reduce ion beam damage.

Comment #15: *What is a compliant epoxy resin?*

Response: Thank you for noticing this typo. What we meant was “commercial epoxy resin”. Correction has been made to the revised manuscript:

Manuscript:

...and embedded in a soft and commercial epoxy resin (EpoFix, Struers Co., Ltd.).

Comment #16: *“FIB-SEM (FIE SCIOS-ZEISS Supra55)”: both FIBs were used? For which experiments?*

Response: Both 3D FIB tomography and 2D SEM characterization were carried out by FIB-SEM (FIE SCIOS-ZEISS Supra55). The detail has been added to the manuscript:

Manuscript:

For both 3D FIB tomography and 2D SEM characterization, electrodes were transferred from the glove box to the FIB-SEM (FEI Scios-ZEISS SUPRA[®] 55) using a transfer vessel to avoid any exposure to air.

Comment #17: *There is no Fig 2k.*

Response: Sorry about the mistake, changes have been made accordingly in the manuscript:

Manuscript:

A key finding from Fig. 2d and 2e is that...

Comment #18: *The least to do is to give a value for x in SiO_x . One could wonder if similar results are obtained for different values of x and it is necessary to be able to reproduce exposed results.*

Response: Thank you for your kind suggestions. We have acquired a different SiO_x sample. Inductively coupled plasma optical emission spectroscopy results have shown that this sample has a much higher x value (0.98) compared with the one we used in the article (0.68). Similar SEI layer structures can be observed on this SiO_x after cycling, suggesting this is a universal technique with reproducible results for different samples. Relevant contents have been added to the manuscript:

Manuscript:

...commercial grade SiO_x (x is measured to be 0.68) microparticles with a D50 value of 5.2 μm ...

Similar SEI can be observed on SiO_x particles with a much higher x value (Supplementary Fig. 9).

Elemental analysis. Different SiO_x samples were dissolved in solution, then diluted for inductively coupled plasma optical emission spectroscopy (ICP-OES) measurement (Agilent ICP-OES 725ES).

Supplementary Information:

Supplementary Fig. 9 | Cross-sectional SEM images of a SiO_x ($x=0.98$) electrode after 300th cycles.

REVIEWER COMMENTS

Reviewer #1 (Remarks to the Author):

I would like to thank the authors for their detailed responses to all the comments and questions, and the corresponding revisions/additions to the manuscript. Also, it is good to see that the authors assured us of the soundness of the proposed strategy by excluding some undesirable effects (e.g., beam damage), which makes the conclusion more convincing. Therefore, I would like to recommend the publication as is.

Reviewer #2 (Remarks to the Author):

Following my comments, large changes were made to the manuscript (and supplementary). Some of them are very welcome. I must say some answers were not very clear to me. Some statements are clearly opposite to what the authors probably mean (response 1 “Admittedly, we cannot guarantee perfect sample preservation since FIB is indeed a constructive sample preparation method”)! I would also advise a spelling/grammar check.

Responses # 2, 4, 5, 7, 8, 9, 10, 11, 12, 15, 17, 18 are satisfactory even if, in some of them, suppressing a figure is hardly an answer.

Responses # 1. I still have doubts on the FIB measurements. Without cryo conditions, Ga⁺ ions are too energetic for the SEI. The cryo-microtomy seems to show a similar SEI. But I doubt the images were performed at low temperature, so the electron beam could provoke the same kind of images (porous SEI) as those with FIB. I know well silicon electrodes within FIB, but maybe SiO_x ones have a different behaviour.

Response #3: I beg to differ. Ar⁺ etching can change the interpretation of XPS spectra. On survey spectra, the effect cannot be seen so Fig. R3 is not very informative.

Response #6: as expected, with less CA the electrode performance is much worse. The authors state that it basically does not matter and even does agree with their conclusions. I doubt that. But I agree this is not the main result of the study.

Response #13: to me it is still not clear. What is the “ion milling system”? Is that an ion polisher using Argon ions? Or is it still the FIB?? This must be clear since sample preparation is at the core of the publication.

Response #14: it is rather surprising that the authors indicate that they deposited Pt all around the SiO_x particles (see Fig; R7a).

Response #16: to be clearer, FIE does not exist (must be FEI...) and SCIOS is a FIB from FEI not ZEISS.

So please write "FEI SCIOS and ZEISS supra 55". SCIOS for 3D and Supra for 2D?

Reviewer #3 (Remarks to the Author):

General comment

The authors present a study of SEI growth on SiO_x electrodes through FIB-3D visualization. They proposed a model for the SEI growth based on formation on porous SEI layer underneath a "dense SEI" during cycling. At the end of the paper, they proposed to "confine" the electrode using a graphite layer on top of it to restrict the SEI formation which seems to have a positive effect. I am not fully convinced by the mechanism proposed and some of the results are questionable.

Comment 1.

The morphology of the SEI (sample after 300 cycles) after ion milling cross sectioning and cryo-ultramicrotomy (Fig S10) is said to be the same and that beam damage are negligible. I do not fully agree with this answer. A porosity is indeed visible on sample after cryo-ultramicrotomy however it seems less pronounced than the one visible on ion milled sample. As the SEI is most probably composed of a mixture of polymer-like compounds as well as inorganic ones, a differential pulverization between the different compounds in the SEI and even the SiO_x is highly possible. One can then wonder about an effect of the Ga⁺ irradiation on an increase in porosity or even generation of porosity (through gas generation for example which is a classical observation on polymer-like compounds).

Comment 2.

The proposed mechanism is surprising in the sense that the "dense" layer which is progressively formed during cycling, does not crack or break while a porous part often linked to a volume expansion grows underneath. How exactly is the volume expansion due to the formation of an underlying porous structure accommodated without breaking the upper layer as generally observed in silicon-based anode?

Comment 3.

On the EELS results, I am not fully convinced by the mapping allowing the distribution of the carbon from CA-binder and the rest of the inorganic or organic polymer-like compounds. The reference for the CA-binder was taken in a sample already submitted to cycling and to FIB milling so I disagree about taking it as reference for a mapping. The carbon present in the polymer-like compounds in the SEI can also have a similar shape. The analyses of a real reference of CA-binder taken on a fresh sample would have been preferable and the investigation of the valence (plasmon) electron energy loss spectroscopy (VEELS) which can also be used as a fingerprint would have been interesting.

Comment 4.

In Figure S8, at the end of the caption, it is referred to Figure S10 “d-e” but there is no “e” in figure S10.

In figure S10, “b” and “d” are an enlarge view of respectively “a” and “c”, it should be precise in the caption.

In figure S14, it is written “the corresponding EDS mapping of Si and C.” but the carbon map is not presented.

Comment 5.

In my opinion, the author should put the non-etched results as the observations in the global spectra between non-etched and etched samples are really similar. The conclusion will remain the same. However, an XPS spectrum is really dependent of the analyzed area so I would be more moderate about the carbon conclusion.

Point-to-point response to reviewers

Reviewer #1

General Comments: *I would like to thank the authors for their detailed responses to all the comments and questions, and the corresponding revisions/additions to the manuscript. Also, it is good to see that the authors assured us of the soundness of the proposed strategy by excluding some undesirable effects (e.g., beam damage), which makes the conclusion more convincing. Therefore, I would like to recommend the publication as is.*

Response: We are truly thankful to the reviewer for the positive assessment of our work.

Reviewer #2

General Comments: *Following my comments, large changes were made to the manuscript (and supplementary). Some of them are very welcome. I must say some answers were not very clear to me. Some statements are clearly opposite to what the authors probably mean (response 1 “Admittedly, we cannot guarantee perfect sample preservation since FIB is indeed a constructive sample preparation method”)! I would also advise a spelling/grammar check.*

Response: Again, we would like to thank the reviewer for the time and effort taken in reviewing our manuscript. To address the reviewer’s concern, we have carried out additional experiments as demonstrated in the point-to-point response. We are truly thankful for the constructive comments that would significantly improve this work.

Comment #1: *Responses # 2, 4, 5, 7, 8, 9, 10, 11, 12, 15, 17, 18 are satisfactory even if, in some of them, suppressing a figure is hardly an answer.*

Response: We thank the reviewer for the positive comments on our revisions.

Comment #2: *Responses # 1. I still have doubts on the FIB measurements. Without cryo conditions, Ga⁺ ions are too energetic for the SEI. The cryo-microtomy seems to show a similar SEI. But I doubt the images were performed at low temperature, so the electron beam could provoke the same kind of images (porous SEI) as those with FIB. I know well silicon electrodes within FIB, but maybe SiO_x ones have a different behaviour.*

Response: We are truly thankful for the reviewer's insightful comment. Indeed, all SEM images of SiO_x treated by cryo-microtomy were taken with cold field emission SEM without cryo conditions. Also, we agree with the reviewer that both Ga⁺ and electron beam could cause damage to SEI, especially for Si anodes. Compared with SiO_x, Si suffer from much larger volume swings, causing constant destruction-reconstruction of SEI, hence a much thinner and fragile SEI. Based on this, we have no doubt about this technique would severely change the SEI morphology on Si particles. Nevertheless, thanks to the less significant volume changes of SiO_x, the SEI grown on SiO_x undergoes a marked thickening process as we proposed in this work. Since the resulted SEI is way more robust than conventional SEIs reported in most studies, the beam damage to such thick SEI morphology would be much less significant under sub-micron scales for SEM observation. To better illustrate the suitability of FIB-SEM for probing this particular system, the following discussions have been added to the revised manuscript:

Main text:

Additionally, the interference of beam damage to the sample was studied by comparing samples prepared from FIB and cryo-ultramicrotomy (**Supplementary Fig. 10**). Although a slightly higher porosity can be observed in the sample prepared by FIB (which can be attributed to the possible beam damage), both samples exhibit SEIs with a loose interior layer and a dense exterior layer, confirming the suitability of this method for observing SEI with considerable thickness (sub-micron level). In comparison, for anode materials with thin and fragile SEI layers (e.g. Si), both Ga⁺ and electron beams might distort the SEI morphology.

Interestingly, SEI with such thickness has not been observed on Si anodes, which might be attributed to the large volume variation of Si generally leads to particle pulverization and collapse of SEI during its early growing stage, hence no Type II SEI can be formed.

Comment #3: Response #3: I beg to differ. Ar⁺ etching can change the interpretation of XPS spectra. On survey spectra, the effect cannot be seen so Fig. R3 is not very informative.

Response: We thank the reviewer for the comments. We agree that the Ar⁺ etching may bring unnecessary complexity to the XPS spectra. Therefore, we have replaced the spectra with the pristine electrode without Ar⁺ etching:

Supplementary Information:

Supplementary Fig. 7 | Full XPS spectra of the SiO_x electrode **a**, before cycling; **b**, after 1st lithiation; **c**, 1st delithiation; **d**, 300th lithiation and **e**, 300th delithiation.

Comment #4: *Response #6: as expected, with less CA the electrode performance is much worse. The authors state that it basically does not matter and even does agree with their conclusions. I doubt that. But I agree this is not the main result of the study.*

Response: We apologize for the misleading conclusion in our last response. We agree that with less CA, almost all cathode/anode materials will deliver poorer performance, it does matter in terms of battery performance. However, as the reviewer has pointed out, it is difficult to correlate this result with the SEI aging process. Therefore, this general phenomenon cannot be seen as an evidence to either support or oppose our claims, and we choose not to discuss this result in this work. We appreciate the reviewer's kind understanding.

Comment #5: *Response #13: to me it is still not clear. What is the “ion milling system”? Is that an ion polisher using Argon ions? Or is it still the FIB?? This must be clear since sample preparation is at the core of the publication.*

Response: We thank the reviewer for the detailed comments. Yes, it is an ion polisher using Ar ions (as shown in **Fig. R1**).

Fig. R1 | Photo of Gatan Precision Ion Polishing System.

The relevant details have been included in the revised manuscript:

Main text:

Methods

To minimize amorphization and damages caused to the cross section, an ionic polishing step was carried out using an Ar-ion polishing system (Gatan Precision Ion Polishing System, PIPS II Model 695) under 2 keV/30 μ A.

Comment #6: *Response #14: it is rather surprising that the authors indicate that they deposited Pt all around the SiO_x particles (see Fig; R7a).*

Response: We thank the reviewer for the detailed comment. We apologize for not explaining the mechanism of Pt deposition more clearly. In the FIB-SEM chamber, Pt is deposited through thermal evaporation coating. Therefore, gaseous Pt will inevitably adsorb on all exposed surfaces, forming a surrounding Pt layer. The particle we showed in last response letter was not fully embedded in the electrode, hence the seemingly fully covered SiO_x particle. We have included additional descriptions in the revised manuscript:

Main text:

Methods

Areas for observation were pre-coated with ~1 μm of platinum (Pt) via a Pt gun based on thermal evaporation in the FIB-SEM chamber.

Comment #7: *Response #16: to be clearer, FIE does not exist (must be FEI...) and SCIOS is a FIB from FEI not ZEISS. So please write “FEI SCIOS and ZEISS supra 55”. SCIOS for 3D and Supra for 2D?*

Response: We would like to thank the reviewer for pointing this out. The FIB-SEM system is an integrated system consisting of both FEI SCIOS (FIB) and ZEISS SUPRA 55 (SEM). To make the description clearer, the relevant content has been revised as below:

Main text:

Methods

An integrated FIB-SEM system consisting of FEI SCIOS (FIB) and ZEISS SUPRA® 55 (2D SEM) is employed for tomographic imaging.

Reviewer #3

General Comments: *The authors present a study of SEI growth on SiO_x electrodes through FIB-3D visualization. They proposed a model for the SEI growth based on formation on porous SEI layer underneath a “dense SEI” during cycling. At the end of the paper, they proposed to “confine” the electrode using a graphite layer on top of it to restrict the SEI formation which seems to have a positive effect. I am not fully convinced by the mechanism proposed and some of the results are questionable.*

Response: We are grateful for the time and effort taken in reviewing our manuscript. Revisions have been made accordingly to the reviewer’s constructive comments that would significantly improve this work.

Comment #1: *The morphology of the SEI (sample after 300 cycles) after ion milling cross sectioning and cryo-ultramicrotomy (Fig S10) is said to be the same and that beam damage are negligible. I do not fully agree with this answer. A porosity is indeed visible on sample after cryo-ultramicrotomy however it seems less pronounced than the one visible on ion milled sample. As the SEI is most probably composed of a mixture of polymer-like compounds as well as inorganic ones, a differential pulverization between the different compounds in the SEI and even the SiO_x is highly possible. One can then wonder about an effect of the Ga⁺ irradiation on an increase in porosity or even generation of porosity (through gas generation for example which is a classical observation on polymer-like compounds).*

Response: We would like to thank the reviewer for the insightful comments. After careful deliberation, we agree with the reviewer that the beam damage to the SEI, especially the organic components, is not negligible. We apologize for the unsolid claims, and the corresponding revisions have been made to the manuscript:

Main text:

Additionally, the interference of beam damage to the sample was studied by comparing samples prepared from FIB and cryo-ultramicrotomy (**Supplementary Fig. 10**). Although a slightly higher porosity can be observed in the sample prepared by FIB (which can be attributed to the possible beam damage), both samples exhibit SEIs with a loose interior layer and a dense exterior layer, confirming the suitability of this method for observing SEI with considerable thickness (sub-micron level). In comparison, for anode materials with thin and fragile SEI layers (e.g. Si), both Ga⁺ and electron beams might distort the SEI morphology.

Nevertheless, it should be noted that the sample prepared through cryo-ultramicrotomy still shows porous SEI structure in the inner layer, despite the pores are less pronounced than the sample prepared from FIB, and both samples exhibit dense outer SEI layers. This result could be attributed to that the loose inner SEI layer is more susceptible to Ga⁺ beam, hence the slight porosity increase. On the whole, the difference in porosity will not necessarily affect our main conclusion, where two types of SEI are formed on SiO_x – the dense inner layer and the loose inner layer.

More importantly, the reviewer's comments have also brought us a new revelation that since the porosity could be affected by sample preparation methods, it is no longer appropriate to define Type I SEI as "porous". Instead, we believe "loose" and "incompact" are more suitable to describe Type I SEI. Some representative changes in both texts and figures have been made as shown below. We hope our revisions could relieve the reviewer's concerns.

Main text:

...we reveal an exceptionally characteristic SEI microstructure with an **incompact** inner region and a dense outer region...

...there appear to be a boundary developed on the surface of this outer layer and the originally **loose** space between SiO_x particles is filled out...

As the cycle continues, **the loosely structured** Type I SEI not only grows thicker, but also evolves into the dense Type II SEI...

Comment #2: *The proposed mechanism is surprising in the sense that the "dense" layer which is progressively formed during cycling, does not crack or break while a porous part often linked to a volume expansion grows underneath. How exactly is the volume expansion due to the formation of an underlying porous structure accommodated without breaking the upper layer as generally observed in silicon-based anode?*

Response: We thank the reviewer for the insightful comments, which are very helpful for improving this manuscript.

First, it should be noted that being dense is not necessarily equivalent to being rigid. We apologize for using the term “hard shell” to describe Type II SEI, which might cause the confusion. Since SEI layers generally consists both organic and inorganic components, the former facilitates **reasonable resilience to tolerate the small stress caused by the volume change**. Therefore, a thin dense layer could progressively grow into a thick dense layer during long-term cycling. Corresponding changes have been made to avoid further misunderstanding:

Main text:

As the cycle continues, the loosely structured Type I SEI not only grows thicker, but also evolves into the dense Type II SEI, whose morphology remains relatively stable during sequential cycles.

Upon delithiation, the Type I SEI is stretched with the shrinkage of SiO_x without causing structural collapse of Type II SEI; whereas during lithiation, the Type I SEI is compressed between the “dense shell” (Type II SEI) and the expanding SiO_x particle, blurring the boundary between Type I and Type II regions.

It should be noted that due to the existence of soft polymeric SEI components, the outer layer can mechanically withstand the volume changes as most of the stress can be buffered by the inner layer.

Second, **by looking at just one lithiation-delithiation cycle**, the delithiated state is illustrated in **Fig. R1**, where compressed Type I and Type II SEI co-existed.

Fig. R2 | Deformation process of Type I SEI during cycling.

We apologize for not including the compressed Type I in the old version of **Fig. 3c** in the original submission, which might have caused the confusion. After delithiation, the compressed Type I SEI is stretched along with shrinking SiO_x , forming porous Type I. Consequently, the denser Type II SEI will not experience drastic volume change. To avoid further confusion, we have edited Fig. 3 and the corresponding discussions as below:

Thirdly, we understand that for Si anodes, such SEI thickening has not been observed. This is due to the large volume change of Si ($\sim 300\%$) does not allow such process: on the one hand, nanosized Si cannot support such thick SEI; on the other hand, for micro-sized Si particles, even they can avoid pulverization (which is always the case), we speculate that most Type II SEI (e.g. without electrolyte optimization or material modification) cannot be formed as the large volume swing will destroy it at the very beginning. In comparison, exhibiting a moderate volume swing, the surface of SiO_x is suitable for SEI thickening. To better demonstrate significance of this work, we have included the following discussions in the revised manuscript:

Main text:

Interestingly, SEI with such thickness has not been observed on Si anodes, which might be attributed to the large volume variation of Si generally leads to particle pulverization and collapse of SEI during its early growing stage, hence no Type II SEI can be formed.

Comment #3: *On the EELS results, I am not fully convinced by the mapping allowing the distribution of the carbon from CA-binder and the rest of the inorganic or organic polymer-like compounds. The reference for the CA-binder was taken in a sample already submitted to cycling and to FIB milling so I disagree about taking it as reference for a mapping. The carbon present in the polymer-like compounds in the SEI can also have a similar shape. The analyses of a real reference of CA-binder taken on a fresh sample would have been preferable and the investigation of the valence (plasmon) electron energy loss spectroscopy (VEELS) which can also be used as a fingerprint would have been interesting.*

Response: We are truly thankful for the reviewer's constructive comments. We agree that the reference should come from a fresh sample. Therefore, we have taken a new reference from a pristine SiO_x electrode and re-drawn EELS mapping based on the new reference, which does look a bit different. The corresponding revisions are listed below:

Main text:

Moreau and co-workers have also employed low-loss EELS to identify SEI species on nano-sized Si anode.²²

To directly prove this assumption, EELS was used to analyze different areas of an electron transparent FIB lift-out lamellae including a SiO_x particle, its SEI and surrounding CA-binder matrix (**Fig. 4e**). The reference point was sampled from the CA-binder domain in a pristine SiO_x electrode, which exhibits very similar C K-edge spectra with CA-binder domain in the cycled electrode (**Supplementary Fig. 13**).

Next, we quantitatively determine the degree of similarity between signals collected from a selected domain of SEI on a cycled SiO_x (**Fig. 4g**) with the reference. Consequently, an EELS mapping (**Fig. 4h**) of the relative conductive composition concentration can be obtained (see detailed calculation descriptions in **Supplementary Fig. 13**).

Fig. 4h | The planar concentration distribution of conductive agents.

Supplementary Information:

Supplementary Figure 13 | Electron transparent FIB lift-out lamellae of **a**, cycled and **b**, pristine SiO_x particle and its SEI with surrounding CA-binder domain. **c**, EELS spectra of the C K-edge from P1, P2 in the uncycled CA-binder domain and P0 in the CA-binder domain of cycled SiO_x.

Next, we used Point 1 (CA-binder domain) in Supplementary Fig. 13b as the reference (xi) to evaluate...

Indeed, Moreau and co-workers have demonstrated that VEELs would be useful as chemical fingerprints for identifying specific species in SEI of Si anode (*Nano Letters*, **2016**, 16, 7381-7388) and we have cited this work as Ref. 22 in the revised manuscript. Nevertheless, our goal is to quantify the distribution of CA-binder mixture within the SEI, instead of identifying a specific chemical component. Therefore, core-loss EELS might be more powerful in our study. The low-loss data are provided along with the core-loss data below in **Fig. R3**, which seems to be more

complicated to extract the desirable information of CA-binder mixture. Still, we thank the reviewer for broadening our vision, which is very useful in our future componential study of SEI.

Fig. R3 | Electron transparent FIB lift-out lamellae of **a**, cycled and **b**, pristine SiO_x particle and its SEI with surrounding CA-binder domain. **c**, low loss spectra of different points in different electrodes.

Comment #4: In Figure S8, at the end of the caption, it is referred to Figure S10 “d-e” but there is no “e” in figure S10. In figure S10, “b” and “d” are an enlarge view of respectively “a” and “c”, it should be precise in the caption. In figure S14, it is written “the corresponding EDS mapping of Si and C.” but the carbon map is not presented.

Response: We apologize for the confusing figure captions in Supplementary Figure 8. We forgot to delete that sentence. Additionally, the figure caption of Supplementary Figure 10 is corrected correspondingly.

Supplementary Information:

Supplementary Fig. 8 | **a**, The electrode surface was embedded in the resin. **b**, The specimen was gradually polishing with a triple Ar ion-beam milling system. **c**, The polished electrode showed a super-flat top surface. **d-e**, The cross-sectional images of the topmost particles in the electrode at the 300th delithiated state, investigated by SEM and optical microscope, respectively. The scale bar = 10 μm .

As for Supplementary Figure 14, we did include the mapping of carbon. To void confusion, we have rearranged and numbered all figures:

Comment #5: *In my opinion, the author should put the non-etched results as the observations in the global spectra between non-etched and etched samples are really similar. The conclusion will remain the same. However, an XPS spectrum is really dependent of the analyzed area so I would be more moderate about the carbon conclusion.*

Response: We thank the reviewer for the comments. We agree that the Ar⁺ etching may bring unnecessary complexity to the XPS spectra. Therefore, we have replaced the spectra with the pristine electrode without Ar⁺ etching. Moreover, we also agree that the carbon conclusion might be a little hasty, hence it has been deleted from the supplementary information:

Supplementary Information:

Supplementary Fig. 7 | Full XPS spectra of the SiO_x electrode **a**, before cycling; **b**, after 1st lithiation; **c**, 1st delithiation; **d**, 300th lithiation and **e**, 300th delithiation.

REVIEWERS' COMMENTS

Reviewer #3 (Remarks to the Author):

general coments:

I would like to thanks the authors for answering in details the comments and questions raised. I much appreciated the clearer explanation on the mechanism proposed. The authors address the different questions raised and answer them with constructed and convincing answers. They considered the possible drawbacks of the methodology used and amend the text accordingly. I agree that the SEI growth mechanism from SiOx proposed is a plausible explanation in agreement with obtained results.